# Argan (*Argania spinosa* L.) Seed Oil Cake as a Potential Source of Protein-Based Film Matrix for Pharmaco-Cosmetic Applications

**DOI:** 10.3390/ijms23158478

**Published:** 2022-07-30

**Authors:** Seyedeh Fatemeh Mirpoor, Concetta Valeria L. Giosafatto, Loredana Mariniello, Antonella D’Agostino, Maria D’Agostino, Marcella Cammarota, Chiara Schiraldi, Raffaele Porta

**Affiliations:** 1Department of Chemical Sciences, Montesantangelo Campus, University of Naples “Federico II”, via Cintia 4, 80126 Naples, Italy; seyedehfatemeh.mirpoor@unina.it (S.F.M.); loredana.mariniello@unina.it (L.M.); raffaele.porta@unina.it (R.P.); 2Department of Experimental Medicine, Section of Biotechnology and Molecular Biology, University of Campania “Luigi Vanvitelli”, 80138 Naples, Italy; antonella.dagostino@unicampania.it (A.D.); maria.dagostino@unicampania.it (M.D.); marcella.cammarota@unicampania.it (M.C.); chiara.schiraldi@unicampania.it (C.S.)

**Keywords:** argan seed oil, seed oil cake, protein-based film, biomaterials, cosmeceutical sector, food coating

## Abstract

Various different agri-food biomasses might be turned into renewable sources for producing biodegradable and edible plastics, potentially attractive for food, agricultural and cosmeceutical sectors. In this regard, different seeds utilized for edible and non-edible oil extraction give rise to high amounts of organic by-products, known as seed oil cakes (SOCs), potentially able to become protein-rich resources useful for the manufacturing of biodegradable films. This study reports the potential of SOC derived from *Argania spinosa* (argan), a well-known plant containing valuable non-refined oil suitable for food or cosmetic use, to be a promising valuable source for production of a protein-based matrix of biomaterials to be used in the pharmaco-cosmetic sector. Thus, glycerol-plasticized films were prepared by casting and drying using different amounts of argan seed protein concentrate, in the presence of increasing glycerol concentrations, and characterized for their morphological, mechanical, barrier, and hydrophilicity properties. In addition, their antioxidant activity and effects on cell viability and wound healing were investigated. The hydrophobic nature of the argan protein-based films, and their satisfying physicochemical and biological properties, suggest a biorefinery approach for the recycling of argan SOC as valuable raw material for manufacturing new products to be used in the cosmeceutical and food industries.

## 1. Introduction

Oil seeds represent one of the most important crops in the world, because they are energy dense foods, due to their high oil content, and good sources of proteins, fibers, vitamins, and minerals. Oil seed crops are primarily grown for edible oil production, and the fat yields of oil seeds are generally quite high, although variable from crop to crop and depending on the methods of extraction [1,2]. It is worth highlighting that some specific oil seeds have recently attracted particular attention for their additional ‘benefits’. In fact, some vegetable oils resulting from their processing have become important either for their pharmaceutical and cosmetic properties, or for bio-fuel production and other industrial oleo-chemical uses [3]. Nevertheless, oil seed processing leads to the conspicuous production of waste materials. These can, however, be recovered as by-products (hull, meal, oil cakes) useful for integration into animal as well as human diets due to their valuable nutrient compositions [1]. In particular, the term ‘seed oil cake’ (SOC) indicates the mass deriving from the oil extraction process from seeds that contains a high protein content together with a variable amount of fat still present. These SOCs are voluminous food-grade industrial by-products, and novel biotechnological approaches are needed to valorize the different industrial co-streams containing high amounts not only of oil but also of SOC. Even though the residues of oilseed crops are known to be extremely rich in valuable ingredients, containing, besides proteins, also fiber and various bioactive compounds, these by-products are so far mostly consumed as animal feed supplementation, food additives, or fertilizers in their production areas [2]. Nevertheless, the relatively high protein content and its inexpensive cost make SOC an adequate resource for the development of innovative bioplastic materials [1,4]. In fact, when SOCs are used to raise the protein level of animal diets, they must be complemented by legume seeds, as well as animal by-products and synthetic amino acids, because they are usually deficient in certain essential amino-acids. Therefore, among the biopolymers derived from renewable resources, SOC-extracted proteins, without or after purification, could be a potential raw material for bioplastic products, since they are abundant, biodegradable and inexpensive [2].

*Argania spinosa* L., known as argan, is a slow growing thorny tree belonging to the *Sapotaceae* family. Argan is an endemic plant of Morocco [5], and is also cultivated in desert and semi-desert regions, inclusing the south-west of Algeria, and in the deserts of South Africa, Spain, Kuwait, Mexico, Tunisia, and Israel [6,7]. Argan plays a significant environmental role, thanks to its ability to combat desert progression through its deep root system that protects the soil from water and wind erosion [8,9].

The plant fruit (2–4 cm long and 1.5–3 cm thick) is an oval drupe with a sweet-smelling but unpleasantly flavored layer of pulpy pericarp surrounded by a thick and bitter peel. It contains an extremely hard nut, inside which there are up to three seeds [10], commercially called argan almonds. The argan seed oil (ASO), with its golden color and characteristic nutty flavor, has long been recognized as a strategic product capable of bringing more income to the Moroccan population. The average content of ASO, generally extracted by a semi-industrial mechanical process, varies between 3.0% and 3.7% of dry fruit weight. However, although the average kernel oil content is around 47% [11], its commercial distributed in Europe and North America has been remarkably expensive because of the extremely low extraction yields (2 to 3.2 kg of oil from 100 kg of dried fruit). ASO is rich in unsaturated fatty acids (80%; oleic, linoleic, and polyunsaturated fatty acids), several bioactive molecules (mainly tocopherols and phytosterols), and a large number of volatile chemical compounds which are responsible for its aroma [8]. Although it is considered one of the most nutritious oils, and thus primarily used as a flavoring ingredient added to a variety of cooked and raw foods, ASO has also been described as a therapeutic agent to prevent cardiovascular risks, by hindering hypercholesterolemia and atherosclerosis [12]. It acts as a choleretic and hepatoprotective [13], and can delay skin aging when used as a cosmetic ingredient [14]. Consequently, there are different types of ASOs on the market for food, pharmaceutic, or cosmetic use. Cosmetic ASO is generally applied to the skin or hair, alone or in more complex preparations, and also as a moisturizer to treat devitalized skin [8,15].

Furthermore, particular attention deserves to be paid to the material that remains after the pressing of the argan seeds to extract the oil. The mechanical press method yields approximately 43% ASO, and the dry residue, which is called argan seed cake, has been reported to contain 48.4% protein, 17.6% fiber, 18.9% fat, significant levels of potassium (10.4 g/kg), calcium (6.9 g/kg), phosphorus (6.4 g/kg), and magnesium (3.3 g/kg), as well as low levels of ash (3.6%) [16]. Although argan seed cake has been used so far almost exclusively to feed animals [12], it has also been proposed as a natural exfoliant and moisturizer for use as an ingredient in commercial shampoo and in creams to topically treat sprains, or scabies, and for healing wounds. In particular, recent results suggest that argan seed cake has melanogenesis regulatory effects, and that its ethanol extract might be used as a component of natural whitening products and as pharmacological agent for the treatment of hyperpigmentation disorders [17,18].

The topic of the present study is the investigation of the possible exploitation of argan seed proteins as biopolymers to obtain biocompatible hydrocolloid films, mainly for potential pharmaco-cosmetic use. In view of this, the manufacturing procedure of these new materials and their main physicochemical and biological characterization were carried out using argan seed protein concentrates (ASPC), and are described here for the first time.

## 2. Results and Discussion

### 2.1. Argan Seed Protein Concentrate Preparation and Characterization

Argan seeds are a good source of proteins, which account for about 50% of the oil cake composition. Therefore, a preliminary study was carried out on ASPC obtained after oil removal from seed flour by means of a Soxhlet apparatus using *n*-hexane. Protein determination of ASPC, prepared from the defatted argan seed flour following acidic/alkaline treatment, indicated that the sample contained 55% protein and that the extracted proteins had molecular masses ranging from 15 to 50 kDa (Figure 1). Moreover, three successive treatments of ASPC (ratio 1:3 *w/w*) with n-hexane showed a progressive decrease in its weight, certainly due to residual amounts of seed oil present in the sample. Thus, it was calculated that ASPC also contained about 20% lipids quite tightly bound to the hydrophobic regions of the argan seed proteins.

Figure 2 shows that ASPC was positively charged in the acidic pH range below pH 4.5; the detected negative zeta potential was observed to decrease progressively from a value of about −35 mV observed at pH 12 to a 0 mV value just under pH 5.0, as a result of the gradual protonation of carboxyl and amino groups occurring on the protein lateral chains. Figure 2 shows that the Z-average size of the ASPC particles varied during the titration of the sample, and particles of high molecular mass were formed at pH ≤ 6 as a result of aggregation of proteins when the latter were close to their isoelectric point. 

### 2.2. Preparation of Argan Seed Protein Concentrate Film Forming Solutions and of Derived Films

The development of argan-protein-based films began with establishing the appropriate amounts of ASPC and plasticizer for preparing stable film-forming solutions (FFSs) to cast and dry, in order to manufacture handleable films. To evaluate the stability of ASPC-based FFSs containing different concentrations of both protein and glycerol (GLY), zeta potential and particle size measurements were carried out. The data reported in Table 1 indicate that all FFSs containing ASPC amounts between 300 mg and 600 mg were markedly stable, independently of GLY concentrations, as their negative zeta potential values were always lower than −30 mV.

In addition, the FFS particle size measurements showed Z-average values, generally higher at higher concentrations of GLY, between about 300 and 750 nm. Finally, the polydispersity index values detected ranged between 0.5 and 0.9, indicating that particles were in solution in a non-uniform size.

In order to manufacture handleable films, FFSs (25 mL each) containing different amounts of ASPC and different concentrations of GLY (%, *w/w* of ASPC) were prepared at pH 12. All FFSs were cast in Petri dishes and placed in a climatic chamber for 24 h at 25 °C and 45% RH. It is worth noting that 300 mg ASPC was found to be the minimum amount necessary to obtain peelable films, and that 30% GLY was the minimum concentration of plasticizer needed to obtain peelable and handleable films under the selected experimental conditions. Moreover, it deserves to be highlighted that FFSs prepared with ASPC obtained after a second cycle of *n*-hexane (for exhaustively removing all the oil present in the sample), gave rise to brittle unpeelable films if cast with 50% GLY. This result clearly indicates that the presence of certain amounts of argan seed oil in the FFS was necessary for the manufacturing of manipulable films that can be defined bio-composites.

### 2.3. Characterization of Argan Seed Protein Concentrate-Based Films

#### 2.3.1. Film Morphology

The obtained material was a yellowish film much more transparent (opacity value = 2.60 ± 0.56 mm^−1^) than that produced with starch (MaterBi^®^, opacity value = 61.92 ± 3.55 mm^−1^), one of the most widely used bio-based biodegradable materials. It appeared less transparent than low-density polyethylene film (LDPE), a well-known commercial oil-based plastic (opacity value = 1.44 ± 0.04 mm^−1^). Images of typical ASPC-based films prepared in the presence of different GLY concentrations are shown in Figure 3. SEM analysis of the films produced in the presence of 50% GLY shows that their surfaces were smooth and homogeneous at both magnitudes analyzed (Figure 4A,B). As far as the film cross-sections are concerned (Figure 4C,D), the visible irregularities and cracks were probably due to the presence of significant amounts of oil remaining in the ASPC, and thus also in the derived film matrix. Similar results were reported by Delgado et al. [19] who prepared films derived from rapeseed protein concentrate.

#### 2.3.2. Film Mechanical Properties

Figure 5 reports the values of the thickness and of the mechanical properties of ASPC-based films prepared from FFSs containing different amounts of ASPC (300–800 mg) and three different GLY concentrations (30, 40 and 50%). It should be noted that only the increasing amount of ASPC was able to increase the film thickness. Moreover, the films prepared using 600 mg of ASPC and 50% of GLY exhibited the highest elongation at break (EB) (~70–90%) and the lowest Young’s modulus (YM) values (~15 MPa) of all the films analyzed. These films also showed acceptable tensile strength (TS) values, similar to those observed in films prepared with lower and higher ASPC amounts (~0.3–0.5 MPa), and not much lower than the maximum TS value that was observed (~1.2 MPa) in the films prepared with 600 mg ASPC and the minimum GLY concentration (30%). It should be noted that the mechanical properties of the petroleum-based plastic LDPE investigated under the same experimental conditions were quite far from those of the ASPC-based materials, the former being very extensible (EB equal to 150%) and more resistant and stiff (TS and YM equal to 6 MPa and 200 MPa, respectively) (data not shown). On the other hand, the bio-based Mater Bi^®^, a starch-based polymer, was found to be even more flexible, with an EB of 317%, and more mechanically resistant, showing a TS of 18.4 MPa and YM of 42.9 MPa (data not shown).

Based on our results and focusing on a possible application of ASPC materials in the cosmeceutical sector, we decided to further characterize the films manufactured, under the experimental conditions described, with 600 mg ASPC and 50% GLY.

#### 2.3.3. Film Moisture Content and Uptake

ASPC-based films were also analyzed for their moisture content and uptake ability, as these features are important for different possible applications, particularly when the films are used under conditions in which water activity is high or when the film has to act as a wound dressing [20,21]. For example, a high moisture content in the coating material considerably limits its use for different kinds of applications including food packaging or in the pharma-cosmetic sector. The results reported in Table 2 show that neither moisture content of nor uptake by the analyzed ASPC-based films were markedly changed by increasing ASPC or GLY amounts in the film matrix. Thus, we decided to continue the characterization of the films manufactured with 600 mg ASPC and 50% GLY, which exhibited the most interesting mechanical properties.

#### 2.3.4. Film Barrier Properties

The ability of ASPC-based films to act as barriers to WV and gases (CO_2_ and O_2_) was studied, and the determined permeability values were compared to those exhibited by two well-known commercial materials, namely Mater Bi^®^, a biodegradable plastic derived from starch, and LDPE, a traditional plastic of petroleum origin. The results reported in Table 3 indicate that the WV permeability of ASPC-based films was much higher (about 100 times) than that exhibited by LDPE, and was only a little lower than that of Mater Bi^®^. Meanwhile, the barrier effect of ASPC-films toward CO_2_ and O_2_ was markedly higher than that exhibited by either of the two commercial plastics tested (more than 10 and 30 times toward CO_2,_ and more than 30 and 180 times toward O_2_, compared with Mater Bi^®^ and LDPE, respectively) [21].

#### 2.3.5. Film Contact Angle and Antioxidant Activity

The ASPC preparation was shown to contain a significant amount of ASO (20%), and the derived film can be defined as a composite film. Its contact angle and antioxidant activity were also considered worthy of investigation (Table 4). The contact angle value of the films derived from FFSs containing 600 mg ASPC and 50% GLY was much higher than that exhibited by other seed-protein concentrate-based films, such as those derived from hemp (HSPC) and cardoon (CSPC) proteins [20,21], and quite similar to that measured after 30 s when analyzing the LDPE films (Table 4). These findings strongly suggest that the significant presence of fat components in the argan protein-based film matrix led to an increase in the hydrophobicity of the film surface, that might enable some industrial applications that are not possible for most protein-based films.

Furthermore, ASPC-based films were found to possess significant antioxidant activity (12.37 ± 1.33%), probably due to their high fatty acid content that characterizes these films as natural bio-composites. Therefore, since these peculiar features suggest potential application of ASPC-based films as bio-active materials, further biological properties regarding their bio-compatibility were investigated.

#### 2.3.6. Film Biocompatibility and Effect on Wound Healing

A cytotoxicity assay was used to evaluate the biocompatibility of increasing amounts (from 0.1 to 5.0 mg) of ASPC-based material immersed in 1.0 mL of DMEM. The results reported in Figure 6 reveal that a high viability of HDF-hTERT was retained when using concentrations of 0.1 mg/mL ASPC-based films, whereas less than 5% of cellular viability normalized over control samples (CTR) was observed when 5.0 mg/mL of the ASCP-based films were present in the DMEM. ASPC-based films were also tested for their wound healing ability; Figure 7 shows a representative field of view of wound closure, in which it was evident that the medium conditioned with a lower amount of film (0.1 mg/mL) was already able to promote slight wound healing. Conversely, the direct addition of the films onto the scratched monolayers hampered natural reparation. Therefore, the biological outcome evidence of the scratch test followed by time lapse videomicroscopy suggested that the ASPC-based films are potentially exploitable in cosmeceutical field, and perhaps even in medical devices aimed at skin repair. These results were confirmed by quantitative analysis in which wound closure was analyzed at representative times. In fact, wound closure occurred slightly faster when 0.1 mg/mL ASCP-based film was present in the medium (20 ± 3% of closure after 12 h; 63 ± 4% after 24 h), compared with the control sample assayed in the absence of film (4 ± 3% of closure after 12 h; 46 ± 11% after 24 h).

## 3. Materials and Methods

### 3.1. Materials

Argan seeds were purchased from a local market in Marrakech (Morocco). Chemicals for electrophoresis were from Bio-Rad (Segrate, Milano, Italy). The reagents for cell viability testing, Dulbecco’s modified Eagle’s medium (DMEM), fetal bovine serum (FBS), penicillin, streptomycin, and fungizone, were purchased from M&M Biotech (Naples, Italy). Sodium hydroxide, hydrochloric acid, and glycerol (GLY) were purchased from Merck Chemical Company (Darmstadt, Germany). The 2,2-diphenyl-1-picrylhydrazyl (DPPH) and N-hexane (99%) were purchased from Sigma Chemical Co. (St. Louis, MO, USA). MaterBi^®^ and low density polyethylene (LDPE) films were from local market shopping bags (Naples, Italy). All other chemicals and solvents used in this study were of analytical grade unless specified.

### 3.2. Preparation of Argan Seed Protein Concentrate

In order to obtain ASPC, argan seeds were ground in a miller (Retsh GH 200) for 3 min, then the oil was extracted with *n*-hexane (ratio 1:3 *w/w*) by means of a Soxhlet apparatus. The derived samples were kept in an oven at 45 °C overnight to allow the *n*-hexane to evaporate. The final ASPCs were obtained by dissolving the dry samples in distilled water (100 mg/mL) and stirring the solutions for 15 min. Then, the pH was adjusted to pH 11 using NaOH 1 N. After 1 h, the samples were centrifuged for 20 min at 10,000× *g* at 4 °C, then the supernatants were collected and adjusted to pH 5.4 to let the proteins precipitate. The solutions were centrifuged again before the pellets were collected and placed in a cabin dryer at 25 °C and 45% relative humidity (RH) for 1 day. ASPC protein content was determined by measuring the nitrogen content of the material and multiplying that value by the factor 6.25 [21,22].

ASPC oil content was determined by submitting ASPC to three successive *n*-hexane extraction steps (ratio 1:3 *w/w*) using the Soxhlet apparatus, and keeping the samples in the oven at 45 °C overnight to let the *n*-hexane evaporate. Since ASPC weight remained constant after the second and the third Soxhlet cycles, it was considered conceivable that all the free oil components had been extracted during the first (16% extracted oil) and the second cycle (4% extracted oil). Therefore, the ASPC oil content was calculated by the difference between the initial weight of dried ASCP and that detected after the second Soxhlet extraction cycle.

### 3.3. Sodium Dodecyl Sulphate Poly Acrylamide Gel Electrophoresis Analysis of Argan Seed Protein Concentrate

For the qualitative protein characterization of ASPC, Sodium Dodecyl Sulphate Poly Acrylamide Gel Electrophoresis (SDS-PAGE) (12%) was performed as described by Laemmli [23]. 25 μL of sample buffer 5× (15 mM of Tris–HCl, pH 6.8, containing 0.5% (*w/v*) SDS, 2.5% (*v/v*) GLY, 200 mM of β-mercaptoethanol, and 0.003% (*w/v*) bromophenol blue) was added to 100 μL of a stock solution of ASPC prepared at a concentration of 2 μg/μL at pH 12. The samples were boiled for 5 min, then increasing amounts of proteins (10, 20, 40 μg) were loaded into the acrylamide gel. Electrophoresis was performed at constant voltage (80 V for 2–3 h) and proteins were revealed with Coomassie Brilliant Blue R250. Bio-Rad Precision Protein Standards were used as molecular weight markers.

### 3.4. Preparation of the Argan Seed Protein Concentrate-Based Film Forming Solutions and of the Derived Films

5 g ASPC was dissolved in 100 mL distilled water, and the pH was adjusted to pH 12 by adding NaOH 1 M and stirring for 1 h. Different ASPC-based FFSs, containing 300, 500, 600 and 800 mg of ASPC in 25 mL of solution, were prepared in the presence of different concentrations of GLY (30%, 40%, 50% *w/w* of ASPC), then stirred for 30 min, poured into Petri dishes, and placed in a climatic chamber at 25 °C and 45% RH for 24 h. These experimental conditions produced handleable and peelable films.

Zeta potential, average size, and polydispersity index values of the different FFSs were determined by using the Zetasizer Nano-ZSP (Malvern). Three independent zeta potential measurements were carried out on each sample of FFS (1 mL) introduced into the measurement vessel. Temperature was set at 25 °C, applied voltage was 200 mV, and duration of each analysis was approximately 10 min. The device used a helium-neon laser of 4 mW output power operating at a fixed wavelength of 633 nm (wavelength of laser red emission). The effect of pH on zeta potential and z-average of ASPC dissolved in water (pH 12.0) was studied by transferring the solution into the autotitrator and adjusting the dispersion pH to different values from pH 12.0 to pH 2.0, adding 1.0, 0.1, or 0.01 N HCL. All the results were reported as mean ± standard deviation.

### 3.5. Film Characterization

#### 3.5.1. Morphological Evaluation

The relative transparencies of ASPC-based film, MaterBi^®^, and low-density polyethylene (LDPE) films were investigated according to the described method [24], based on film opacity determined by the measurement of film absorbance at 600 nm (spectrophotomer UV/Vis SmartSpec 3000 Bio-Rad, Hercules, CA, USA) divided by the film thickness (mm). All the samples were cut into pieces of 1 cm × 3 cm and allowed to perfectly adhere to the wall of the cuvette.

Scanning electron microscope analyses of side surfaces and cross-sections of the ASPC-based films obtained with 600 mg ASPC and 50% GLY were carried out using Supra 40 Zeiss equipment. Films were mounted on stub and were sputter-coated with platinum-palladium (Denton Vacuum Desk V), and micrographs were obtained at EHT = 5.00 kV using an in-lens detector [24].

#### 3.5.2. Mechanical Properties

Before being tested, all dried films were conditioned at 25 °C and 50% RH for 2 h by placing them into a desiccator over a saturated solution of Mg(NO_3_)_2_. After that, the films were cut into 1 cm × 8 cm strips and their thickness was measured using a micrometer, at five points along the entire length of the strip. Then, the average and the respective standard deviation values were calculated. Films’ TS, EB, and YM were determined according to the ASTM method [25] carried out on five specimens for each sample using an Instron universal testing instrument model no. 5543A (Instron Engineering Corp., Norwood, MA, USA). The initial grip separation and crosshead speed were set to 40 mm and 5 mm/min, respectively, and 1 kN cell load was used as indicated by ASTM [25] (1997). The acquisition and elaboration of the data were made using the software BlueHill (New York, NY, USA) (version 2.21).

#### 3.5.3. Moisture Content and Moisture Uptake

Film moisture content analysis was performed to evaluate the mass loss of each film with an initial weight of 0.1 g after 24 h in the oven at 80 °C. The dried samples were equilibrated in CaCl_2_ for 10 min, and then weighed again.

Film moisture content was calculated as:
(1)
$$\mathrm{moisture}\;\mathrm{content}\;(\%)=\frac{Wi-Wf}{Wi}\times 100$$

where *W_i_* is the initial weight of the film and *W_f_* is the film weight after drying.

Film moisture uptake analysis was carried out using the gravimetric method described by Manrich et al. [26]. The analysis was performed by determining the mass of each film after drying at 80 °C for 24 h, and after a further 24 h in a conditioning environment at 50% RH over a saturated solution of Mg(NO_3_)_2,_. The moisture uptake was finally calculated as:
(2)
$$\mathrm{moisture}\;\mathrm{uptake}\;(\%)=\frac{Ws-Wd}{Ws\;}\times 100$$

where *W_s_* and *W_d_* are the weight of swollen and dried films, respectively.

All the experiments were repeated three times.

#### 3.5.4. Permeability and Contact Angle Analyses

Film permeability to O_2_, CO_2_, and water vapor (WV) were detected in duplicate using a Total Perm apparatus (ExtraSolution s.r.l., Pisa, Italy) endowed with barometric compensation, at 50% RH, 25 °C, and 101 kPa for gas permeability (O_2_ and CO_2_), and at 90% RH, 38°C, and 6 kPa for WV permeability [27,28].

Surface hydrophobicity of the films was measured using a homemade water contact analyzer. The strips (2.0 × 2.0 cm) of the different films were placed on the horizontal stage and 10 µL of distilled water was dropped onto the surface of each film. The image of the water drop was captured using a fitted digital microscopic camera, first immediately and again after 30 s of water droplet deposition on the film surface. Five measurements were performed and the average of the contact angle values was calculated [22].

#### 3.5.5. Antioxidant Activity

For the evaluation of the antioxidant activity of ASPC-based films, 20 mg of the film manufactured from FFS containing 600 mg of ASPC and 50% GLY was dissolved in 1.0 mL methanol. Then, 0.1 mL of the obtained solution was mixed with 0.9 mL of methanol DPPH solution (0.005 %, *w/v*), shaken, and left for 30 min in the dark. The absorbance of the collected samples was finally measured at 517 nm by UV/visible spectrophotometer (SmartSpec 3000 Bio-Rad) using methanol as control. All the experiments were carried out in triplicate [29,30].

#### 3.5.6. Cell Culture and Treatments

Immortalized human dermal fibroblasts (HDF-hTERT) were cultured in DMEM containing high glucose, L-glutamine, sodium bicarbonate, and sodium pyruvate, supplemented with 10% FBS, 100 U/mL penicillin, 100 μg/mL streptomycin, and 100 μg/mL antifungal agent. The cells were growth on tissue culture plates (BD Bioscience-Falcon, San Jose, CA, USA) in a humidified atmosphere (95% air and 5% CO_2_, *v/v*) at 37 °C, as previously reported [31,32] with slight modification to evaluate the specific effects of the ASPC-based films.

HDF-hTERT (2 × 10^4^ cells/cm^2^) were seeded on the films by two different methods: (a) the direct method in which different film amounts were added directly into the wells containing previously seeded cells; (b) an indirect method in which the cell medium was preliminarily conditioned for 24 h in the presence of the films and then added to the cells (30–40% cellular confluence in the 3-(4,5-dimethyltiazol-2-yl)-2,5-diphenyltetrazolium bromide (MTT) cell viability test [32], or 100% confluence in the wound healing assay), at a theoretical final concentration ranging from 0.1 to 5.0 mg/mL for the MTT test and only 0.1 and 1 mg/mL for the wound healing assay. All experiments were run using film samples (1–2 mg) suspended in 1.0 mL of culture medium, and the reported amounts are those for the whole film and not the argan proteins contained in it.

#### 3.5.7. Cell Viability

Cytotoxicity of ASPC-based films was assessed by MTT analysis, based on the measurement of the reduction of a yellow tetrazolium salt (3-(4,5 dimethylthiazol-2-yl)-2,5-diphenyltetrazolium bromide), which occurred only in the presence of viable HDF-hTERT cells at confluence of 30–40% [33]. Cells were incubated at 37 °C for 72 h in the presence of different amounts (0.1–5.0 mg/mL) of ASPC-based films. Then, the MTT reagent (0.5 mg/mL in DMEM without FBS) was added and the incubation was prolonged for 3 h at 37 °C. The formazan crystals, presumably directly proportional to the number of viable cells, were solubilized in isopropanol containing HCl 0.1 N and were quantified by recording the changes in absorbance at 570 nm using a UV/visible spectrophotometer (Beckman Coulter, Milan, Italy).

#### 3.5.8. Wound Healing assay and Time Lapse Video Microscopy

Briefly, HDF-hTERT cells (4 × 10^4^ cells/cm^2^) were seeded in 12-well trays until complete cellular confluence was reached. Then, we scratched the confluent cellular monolayers with a sterile tip (Ø = 0.1 mm) and added 1.0 mg/mL of ASPC-based films for both the direct and the indirect method [34].

The *in vitro* cell migration was observed by time lapse video microscopy (TLVM) experiments (OKOLAB, Pozzuoli, Italy) by assembling an inverted microscope (AxioVision200, Zeiss Axiovert 200, Munich, Germany) with a CCD-gray-camera (ORCA ER, Hamamatsu Photonics, Hamamatsu City, Japan) to record the images, and a motorized stage incubator to log the position and maintain the *in vitro* conditions of the cell culture (37 °C, 5% CO_2_ in humidified air). Finally, the custom-tailored software OKO-Vision 4.3 software was employed to follow the overall process and allow image analysis. The TVLM recorded the wound repair for 48–72 h, and enabled selection of the main representative images of the performed experiments. The quantitative analysis of the rate of wound closure was calculated directly by the software as [(Area t0 − Area t)/Area t0] × 100, or, alternatively, by manually tracking the wound area for each image during the process. A minimum of five fields of view were used for each well in order to derive the overall averaged curves of wound closure (%) as a function of time, thus assuring the statistical significance of the experiments.

### 3.6. Statistical Analysis

SPSS19 (Version 19, SPSS Inc., Chicago, IL, USA) software was used for all statistical analyses. One-way analysis of variance (ANOVA) and Duncan’s multiple range tests (*p <* 0.05) were used to determine the significant differences among the samples. All treatments were analyzed in triplicate.

## 4. Conclusions

Bio-composite protein-based films were prepared from the protein concentrate obtained from argan seeds, by casting and drying following the extraction of most of the oil content. The effects of protein concentration and percentage of GLY, used as plasticizer, were studied to determine the best experimental conditions to produce peelable and handleable films with satisfying mechanical, barrier, and hydrophilicity features. The bioactivity of the films with the most attracting physicochemical properties was analyzed, i.e., those manufactured with 600 mg of ASPC and containing 50% GLY. The films showed antioxidant activity and were also able to act as wound dressings, as demonstrated by cell viability experiments and scratch assay. These findings confirm the validity of the strategy that considers seed oil processed cake as a valuable protein-based renewable source for production of biomaterials. Like argan SOC, ASCP retains, in addition to proteins, residues of argan oil, a well-known and valuable component of numerous food additives and of widely commercialized cosmeceutical products. Therefore, argan SOC might represent a potential feedstock to give rise to hydrocolloid films exploitable as dermo-cosmetic devices, as well as food coatings, in the context of a biorefinery where, together with the seed oil, argan by-products can be turned in commodities of higher value.

## Figures and Tables

**Figure 1 ijms-23-08478-f001:**
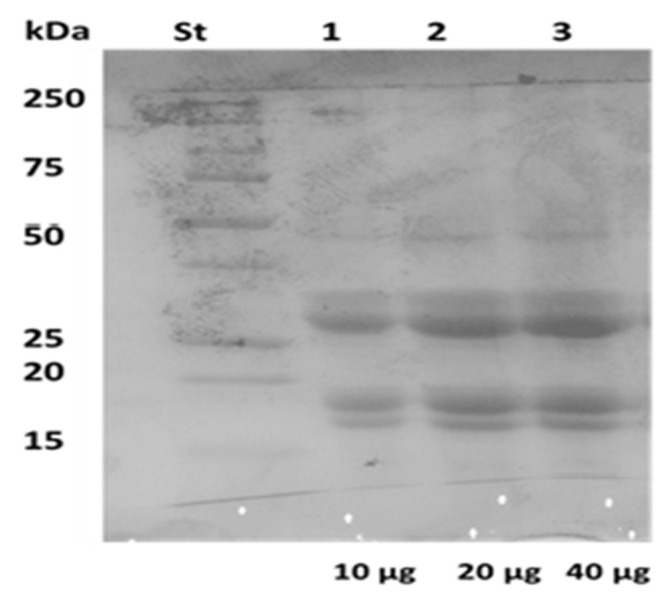
SDS-PAGE (12%) profile of different amounts of argan seed protein concentrate (ASPC).

**Figure 2 ijms-23-08478-f002:**
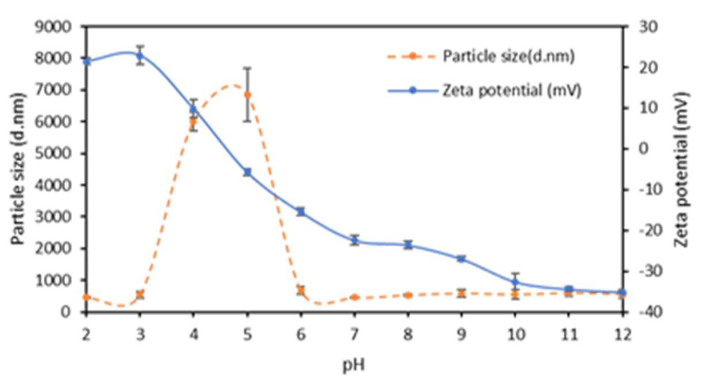
Zeta potential and particle size detected during titration of argan seed protein concentrate. Further experimental details are given in the text.

**Figure 3 ijms-23-08478-f003:**
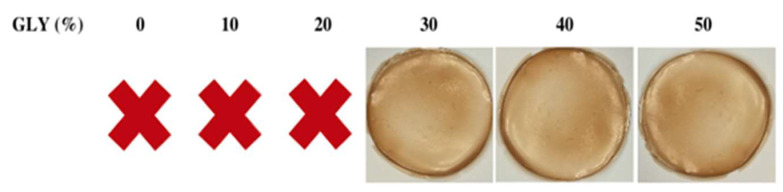
Films obtained at pH 12 using 600 mg argan seed protein concentrate in the presence of different concentrations of glycerol (GLY). No manipulable films (X).

**Figure 4 ijms-23-08478-f004:**
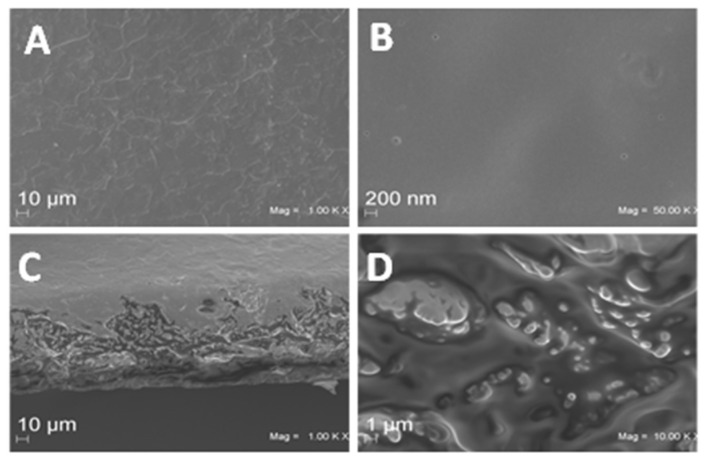
SEM images of argan seed protein concentrate (ASPC)-based films prepared by using 600 mg ASPC in the presence of 50% GLY. Film surfaces (**A**,**B**) and cross-sections (**C**,**D**) were observed at different magnifications: (**A**) 1000×; (**B**) 50,000×, (**C**) 1000×, and (**D**) 10,000×. Further experimental details are given in the text.

**Figure 5 ijms-23-08478-f005:**
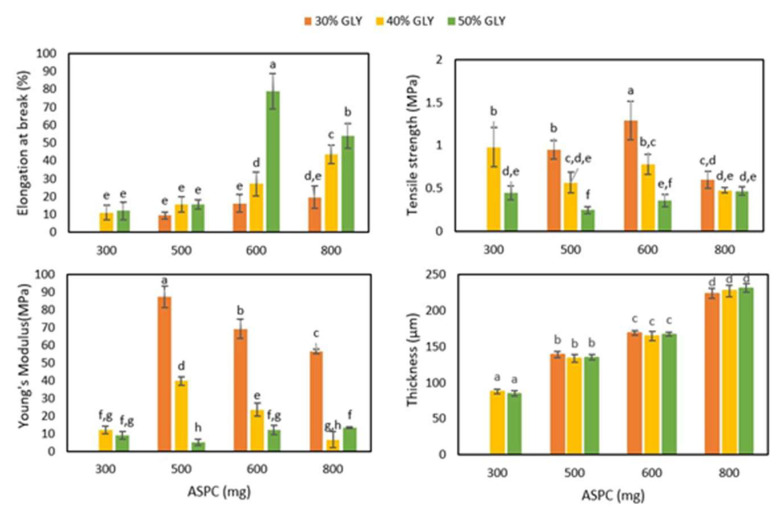
Thickness and mechanical properties of argan seed protein concentrate (ASPC)-based films prepared with film-forming solutions containing the indicated ASPC amounts and glycerol (GLY) concentrations. Different small letters (a–h) indicate significant differences among the values reported (*p* < 0.05). Further experimental details are described in the text.

**Figure 6 ijms-23-08478-f006:**
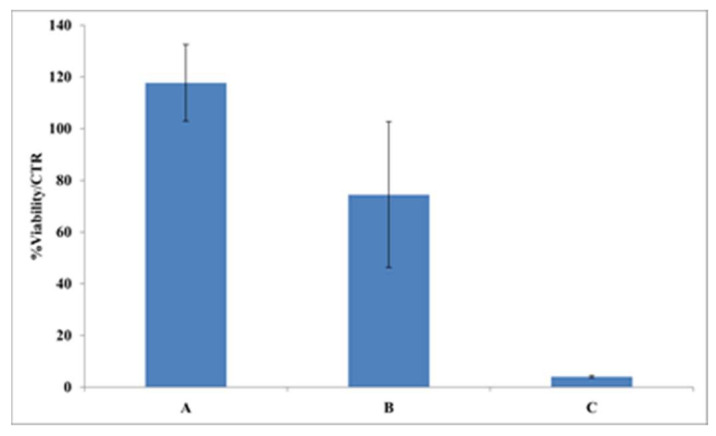
Effects of different concentrations (A, 0.1 mg/mL; B, 1.0 mg/mL; C, 5.0 mg/mL) of argan seed protein concentrate (ASPC)-based films present in the Dulbecco’s modified Eagle’s medium (DMEM) on the viability of immortalized human dermal fibroblast (HDF-hTERT) assayed directly. The absorbance evaluation was carried out after 72 h of treatment. Further experimental details are given in the text.

**Figure 7 ijms-23-08478-f007:**
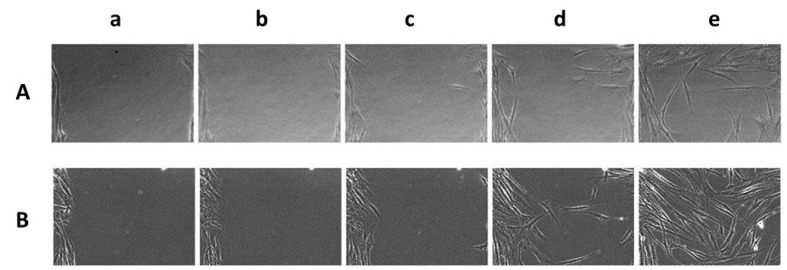
Representative micrograph images of the time course (a, at time 0; b, after 6 h; c, after 12 h; d, after 24 h; e, after 48 h) of the scratch assays of the immortalized human dermal fibroblast (HDF-hTERT), treated in the absence (**A**) or presence (**B**) of 0.1 mg/mL of ASPC-based film, immediately after the scratches.

**Table 1 ijms-23-08478-t001:** Zeta potential, Z-average, and polydispersity index values of film-forming solutions containing increasing amounts of argan seed protein concentrate (ASPC) and glycerol (GLY). Different small letters (a–g) indicate significant differences among the values reported in each column (*p* < 0.05). Further experimental details are given in the text.

ASPC (mg)	GLY (%)	Zeta potential (mV)	Z-Average (nm)	Polydispersity Index
**300**	30	−34.8 ± 1.9 ^a^	410 ± 42 ^d^	0.75 ± 0.02 ^b,c^
	40	−30.0 ± 2.0 ^a^	519 ± 45 ^c^	0.78 ± 0.06 ^b,c^
	50	−34.0 ± 2.0 ^a^	726 ± 10 ^a^	0.90 ± 0.08 ^a^
**500**	30	−36.2 ± 4.9 ^a^	302 ± 11 ^e^	0.69 ± 0.07 ^d,e^
	40	−34.8 ± 3.7 ^a^	320 ± 14 ^e^	0.52 ± 0.02 ^f^
	50	−35.3 ± 3.6 ^a^	394 ± 13 ^e^	0.58 ± 0.02 ^f,g^
**600**	30	−35.9 ± 5.8 ^a^	327 ± 5 ^e^	0.65 ± 0.02 ^e,f^
	40	−35.6 ± 5.7 ^a^	339 ± 15 ^e^	0.66 ± 0.01 ^d,e^
	50	−34.9 ± 5.0 ^a^	579 ± 17 ^b^	0.76 ± 0.08 ^b,c^
**800**	30	−20.3 ± 1.3 ^a^	495 ± 11 ^c^	0.78 ± 0.02 ^b,c^
	40	−20.8 ± 0.9 ^a^	488 ± 26 ^c^	0.75 ± 0.05 ^b,c^
	50	−20.3 ± 0.1 ^a^	590 ± 41 ^b^	0.81 ± 0.09 ^b^

**Table 2 ijms-23-08478-t002:** Moisture content and uptake of argan seed protein concentrate (ASPC)-based films prepared with different amounts of ASPC and glycerol (GLY). Different superscript letters (a–d) indicate significant differences among the values reported (*p* < 0.05). Further experimental details are given in the text.

ASPC (mg)	GLY (%)	Moisture Content (%)	Moisture Uptake (%)
**500**	30	10.03 ± 0.31 ^d^	9.16 ± 0.12 ^b^
	40	10.48 ± 0.28 ^c,d^	9.77 ± 1.34 ^b^
	50	10.12 ± 2.77 ^c,d^	9.22 ± 1.01 ^b^
**600**	30	12.11 ± 0.31 ^b,c^	12.74 ± 1.32 ^a^
	40	12.62 ± 0.29 ^b,c^	12.02 ± 0.97 ^a^
	50	12.20 ± 0.70 ^a^	12.23 ± 1.46 ^a^
**800**	30	11.43 ± 0.43 ^b,c,d^	12.99 ± 0.25 ^a^
	40	12.66 ± 1.18 ^a,b^	13.57 ± 1.06 ^a^
	50	12.10 ± 0.30 ^b,c^	12.21 ± 0.41 ^a^

**Table 3 ijms-23-08478-t003:** Water vapor (WV) and gas (CO_2_ and O_2_) permeability properties of films manufactured with 600 mg of argan seed protein concentrate (ASPC) and 50% glycerol, compared to Mater Bi^®^ and low density polyethylene (LDPE) films. Different superscript letters (a–c) indicate significant differences among the values reported in each column (*p* < 0.05). Further experimental details are given in the text.

Film	WV	CO_2_	O_2_
	(cm^3^ mm m^−2^ d^−1^ kPa^−1^)		
**ASPC**	7.5 ± 0.6 ^a^	0.46 ± 0.08 ^a^	0.02 ± 0.01 ^a^
**Mater Bi^®^**	9.8 ± 0.6 ^b^	5.19 ± 0.60 ^b^	0.69 ± 0.09 ^b^
**LDPE**	0.07 ±0.01 ^c^	13.99 ± 1.08 ^c^	3.79 ± 0.80 ^c^

**Table 4 ijms-23-08478-t004:** Contact angle values of argan seed protein concentrate (ASPC)-based films compared to those prepared with hemp seed protein (HSPC) and cardoon seed protein (CSPC) concentrates, and to Mater Bi^®^ and low density polyethylene (LDPE) films. Different superscript letters (a–e) indicate significant differences among the values reported in each column (*p* < 0.05). Further experimental details are given in the text.

Film	Contact angle (*θ*)	
	At 0 s	After 30 s
**ASPC**	68.26 ± 1.16 ^b^	66.26 ± 1.15 ^b^
**HSPC**	32.15 ± 1.45 ^d^	30.87 ± 1.12 ^d^
**CSPC**	20.76 ± 1.39 ^e^	12.87 ± 1.89 ^e^
**Mater Bi^®^**	64.20 ± 2.03 ^c^	53.85 ± 0.07 ^c^
**LDPE**	70.66 ± 1.35 ^a^	69.75 ± 0.48 ^a^

## Data Availability

The data presented in this study are available on request from the corresponding author.

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
