# Peer review of "Argan (Argania spinosa L.) Seed Oil Cake as a Potential Source of Protein-Based Film Matrix for Pharmaco-Cosmetic Applications"

_ijms, 2022, doi:10.3390/ijms23158478_

Round 1

Reviewer 1 Report

The article titled “Argan (Argania spinosa L.) seed oil cake as potential source of protein-based film matrix for pharmaco-cosmetic applications” is a very interesting document, and by the topic, it deserves being considered. However, some concerns must be better explained and/or simply informed before recommending the publication of the article. This reviewer enjoyed very much the article but has detected that the authors only compare the properties obtained with other reference polymers when one is an advantage respecting the other bio-based polymer compared.

Just to mention that the introduction section is superb.

Under these lines, some recommendations are listed:

·         In the abstract section the following is written: “Among the numerous bio-based plastics proposed as alternatives of conventional oil-derived polluting plastics, one potential option to pursue is the exploitation of agri-food wastes as valuable feedstocks for their production”.  Since this reviewer's expertise is focused on polymer science and technology, this sentence is bold. Oil-based polymers are not pollutant by themselves (in most cases are inert), but by the way they are discarded in a fool society, besides being a way to use the 4% of global crude oil that become a refinery residue once obtained the different fractions useful for combustible, lubricants, and so on. In other words, the production of “oil-based plastics” is a way to recycle the refinery byproducts and feedstocks. In the same way, the so-called biobased polymers in general are far from being eco-friendly at all. So, please, rewrite the sentence to be more scientifically precise, avoiding propaganda sentences like yours.  Note that the high QUALITY OF YOUR PAPER IS BEYOND THIS OPINION CONSIDERATION.

·         The authors only compare the properties obtained with other reference polymers when the one is an advantage, respecting the other bio-based polymer compared.  So, this reviewer wonders why no information about other properties of Materbi® or LDPE is provided. Any convincing reason for it?

·         The SEM images must be better exploited. In this case, information about why just the one with 50% GLY is provided and not the others. Note that material with 50% of plasticizer is a great amount, and the ideal would be to obtain a material with less of this. This reviewer wonders about the absence of images of at least the one with the lower content in GLY able to be handled.

·         The mechanical properties section includes the modulus, and the elongation and strength at break. This reviewer wonders why the authors have not included these values at yield when these are critical in terms of the ultimate properties of a plastic material. At this point, the inclusion of the tensile curves rather than just the bar diagram must be mandatory, at least as supplementary material.

·         The testing rate used (5 mm/min) is excessive for modulus calculations. The recommended in the entire standard is related to the dimensions of the sample, and in your case, it must be much below 0.2 mm/min. Nevertheless, since plenty of universal testing machines are not able to reach such a testing rate, most standards recommend 1mm/min for modulus to be robust.

·         No information about the use of not of extensometer is provided.  Please, supply this information.

In the light of the above-mentioned concerns, and attention to the interest of the article, this reviewer would suggest performing a MAJOR REVISION by considering and/or rebuttal of the about mentioned concern convincingly and robustly.

Author Response

Please find attached the replies to your comments that were able to greatly improve our manuscript

Reviewer 2 Report

Authors reported a new bio-composite protein-based film from argan seed oil cakes by a simple casting/drying method. The basic physical and molecular properties of argan seed protein concentrates (ASPC) were investigated. And the authors studied the effect of ASPC and percentage of glycerol (GLY) on the morphological, mechanical, barrier and hydrophilicity properties of GLY-plasticized films as well as the antioxidant activity and biocompatibility for the potential applications in cosmeceutical and food fields. Overall, this manuscript gave an exhausted investigation on the ASPC films. However, the writing of this paper should be polished, especially for the explanation on the experiment results. So, this manuscript should be carefully revised before publication. Some comments are below:

1. In the Introduction section, there are too many paragraphs to introduce what Argan is, and what the argan seed oil is used for, but the current progress in the preparation and applications of APCS-based films or other protein-based films are less elucidated. So, please tell readers more details about the applications of argan seed oil cakes or other seed oil cakes in any fields.

2. The writing of this manuscript should be improved, and the expression is confusing. E.g., Line 110 - 113, 161-163,

3. Some abbreviations are not given full definition when they appear first in the text, e.g., “FFSs”, “GLY”, etc.

4. The alphabets as annotations in all Tables and Figure 5 are not elucidated in the text.

5. The quality of Figure 7 should be improved.

Author Response

(The authors gave the same response as above.)

Round 2

Reviewer 1 Report

The authors have answered all the concerns convincingly. Consequently, the recommendation is to accept the paper in its actual state.

Reviewer 2 Report

The manuscript has been well revised. It can be accepted in the present form.